# Xerophilic Aspergillaceae Dominate the Communities of Culturable Fungi in the Mound Nests of the Western Thatching Ant (*Formica obscuripes*)

**DOI:** 10.3390/jof10110735

**Published:** 2024-10-23

**Authors:** Rachelle M. Gross, Courtney L. Geer, Jillian D. Perreaux, Amin Maharaj, Susan Du, James A. Scott, Wendy A. Untereiner

**Affiliations:** 1Department of Biology, Brandon University, 270-18th Street, Brandon, MB R7A 6A9, Canada; rachelle_gross@outlook.com (R.M.G.); cgeer1010@gmail.com (C.L.G.); jillian_perreaux33@hotmail.com (J.D.P.); 2Sporometrics Inc., 219 Dufferin Street, Suite 20-C, Toronto, ON M6K 1Y9, Canada; amaharaj@sporometrics.com (A.M.); sdu@sporometrics.com (S.D.); 3Division of Occupational and Environmental Health, Dalla Lana School of Public Health, University of Toronto, 223 College Street, Toronto, ON M5T 1R4, Canada; james.scott@utoronto.ca

**Keywords:** *Aspergillus*, available soil nutrients, low-water-activity habitats, *Penicillium*, *Pseudogymnoascus*, mound-building ants, soil-inhabiting fungi, *Talaromyces*

## Abstract

The nests of mound-building ants are unexplored reservoirs of fungal diversity. A previous assessment of this diversity in the nests of *Formica ulkei* suggested that water availability may be a determinant of the composition of this mycota. To investigate this question, we recovered 3594 isolates of filamentous Ascomycota from the nests of *Formica obscuripes* and adjacent, non-nest sites, employing Dichloran Rose Bengal agar (DRBA), Dichloran Rose Bengal agar containing glycerol (DRBAG), and malt extract agar containing sucrose (MEA20S). Higher numbers of fungi were isolated from the tops of mounds than from within mounds and non-mound sites. Mound nest soils were dominated by members of the family Aspergillaceae, and up to 50% of the colonies isolated on DRBAG belonged to the genus *Aspergillus*. *Pseudogymnoascus pannorum* and species of *Talaromyces* were also present in higher numbers in mound soils. Species of *Penicillium* were more abundant in non-nest soils, where they accounted for over 66% of isolates on DRBA. All Aspergillaceae assessed for xerotolerance on a medium augmented with glycerol or sucrose were xerophilic. These results, and our observation that the nests of *F. obscuripes* are low-water environments, indicate that water availability influences the structure of the fungal communities in these animal-modified habitats.

## 1. Introduction

Mound-building ants (Hymenoptera, Formicidae) build conspicuous above-ground nests composed of excavated soil and organic material in the grasslands and forests of temperate and boreal regions of North America and Eurasia. The organic component of nest mounds, which consists of grass stems, leaf fragments, mosses, conifer needles, and twigs collected by the ants from the surrounding environment, is typically present as a surface thatch that maintains a higher temperature within the nest than in the surrounding soil in the early spring and prevents nests from overheating during the warmer parts of the year [1,2,3,4,5,6,7]. In prairie–forest edge ecosystems, this thatch also regulates moisture levels within nests [1,4].

Mound construction dramatically alters the chemical and physical characteristics of soils and in turn influences the composition and size of the microbial communities in nests. Mound nests differ from surrounding soils in pH, porosity, moisture level, organic matter content, and nutrient availability [3,8,9], and harbor communities of bacteria and fungi that differ markedly from those of non-nest soils [6,10,11,12,13]. Although the impacts of ant-mediated changes on the soil biota are not well understood, differences in acidity, moisture, and temperature have been hypothesized to have the greatest influence on the composition of microbial communities in mound nests [3,6,9,10].

A previous culture-based assessment of the diversity of fungi in the mound nests of *Formica ulkei* revealed that this mycota was dominated by species of *Aspergillus* (Aspergillaceae) [10], a genus isolated commonly from xeric environments and materials with low-water activities [14,15]. In contrast, non-nest soils contained large populations of *Penicillium* and Ascomycota known to be less tolerant of lower water availability. The dominance of species of *Aspergillus* in the nest mounds of *F. ulkei*, their paucity in non-mound soils, and the observation that the moisture content of the nest mounds of this ant species is lower than adjacent soils [1] suggested that water availability may be a major determinant of the composition of this mycota [10].

The xerophilic mycota of indoor and natural environments have been explored in several recent studies [16,17,18,19], but none have examined the potential impact of water availability (i.e., water activity or a_w_, as inferred by measuring equilibrium relative humidity) on the distribution and abundance of fungi in animal-modified habitats. Here, we report the results of a study undertaken to (i) confirm that the community of culturable, filamentous Ascomycota from the mound nests of the western thatching ant (*Formica obscuripes*) differs from the assemblage of fungi in adjacent, non-nest soils; (ii) test the prediction that soils from nests have a lower water activity than those from non-mound sites; (iii) evaluate if mound nests harbor greater numbers and a higher diversity of xerophilic fungi; and iv) determine the xerotolerances of selected isolates of the most abundant taxa recovered from mound nests (i.e., species of *Aspergillus*, *Penicillium*, *Pseudogymnoascus*, and *Talaromyces*). Although various terms have been employed to describe and distinguish between fungi that prefer, tolerate, or require reduced water activity environments (e.g., extreme/moderate/obligate xerophile, mesophilic, osmophilic, xerotolerant) [20,21,22,23,24], we follow Pitt [25] in defining a xerophilic fungus as a species capable of growth, under at least one set of environmental conditions, at a water activity below 0.85.

## 2. Materials and Methods

Soils from three mound nests of *Formica obscuripes* located in the Brandon Hills Wildlife Management Area in southwestern Manitoba, Canada (UTM coordinates of mound 1 = 14N 434487 5508358; mound 2 = 14N 434504 5508369; mound 3 = 14N 434513 5508352), were sampled on 11 and 19 June 2020. Soils were collected with new (unused) plastic spoons from the uppermost 3 cm of mounds (M) and from within nests to a depth of 15–30 cm below the tops of the mounds (L). Non-mound soils (N) were collected 1 m SW of each mound to a depth of 3 cm using a soil core sampler that was cleaned between sites, as described in Duff et al. [10]. All soils were placed into separate, unused plastic ziplock freezer bags, sealed, transported to the laboratory, and processed within 4 h.

### 2.1. Soil Water Activity, Water Content, pH, and Nutrient Analyses

The water activity (a_w_) of soil from each site was determined at 25 °C using a WP4C Water Potential Meter (METER Group, Pullman, WA, USA) following the manufacturer’s instructions. Soil water content was determined as described by Forster [26]. The portion of each fresh collection used for dilution plating was sieved sequentially through 40 mm, 20 mm, and 2 mm meshes to remove plant debris. The smallest size fraction was retained for preparing serial dilutions. The remaining soils were emptied into clean aluminum pans, air-dried at room temperature (RT, 18–21 °C) for 5 d, and stored in new freezer bags.

The pH of air-dried soils was measured following Forster [27]. Analyses of soil electrical conductivity [28] (1:2 soil/water mixture), particle size distribution [29] (method 3.2.1), and nutrient content, including total inorganic carbon [30], available phosphate and potassium [31], total carbon by combustion [32], and available nitrate and sulfate [33], were performed by ALS Global (ALS Environmental Laboratory, Saskatoon, SK, Canada).

### 2.2. Isolation and Identification of Fungi

Fresh soils were used within 2 h of sieving to prepare 10-fold serial dilutions in sterile 0.05% water agar ranging from 10^−1^ to 10^−7^. Soil dilutions from 10^−2^ to 10^−7^ were plated in triplicate on Dichloran Rose Bengal agar (DRBA) [34] containing 25 mg L^−1^ Rose Bengal, 2 mg L^−1^ dichloran, and KH_2_PO_4_, rather than K_2_HPO_4_, Dichloran Rose Bengal agar (DRBAG) containing 18% glycerol, and 2% malt extract agar (MEAS20S) [35] containing 20 g L^−1^ Difco malt extract and 200 g L^−1^ sucrose. All media contained 1.5% agar and were supplemented with 50 mg chlortetracycline hydrochloride L^−1^ and 50 mg streptomycin sulfate L^−1^. The a_w_ of each isolation medium was determined at 25 °C from three uninoculated 72-h-old plates using a WP4C Water Potential Meter.

Inoculated plates were incubated at RT for 4–5 d, after which all colonies of filamentous fungi were transferred to modified Leonian’s agar (MLA) [36] containing 1.5% agar, incubated at RT, and identified to a genus and/or species based on cultural and micromorphological characteristics. Members of the genera *Aspergillus*, *Penicillium*, and *Talaromyces* were identified to species using cultures grown on creatine sucrose agar, Czapek yeast agar, dichloran glycerol agar (DG18), malt extract agar (MEA), and yeast extract agar following Samson et al. [35,37]. The identification of species of *Cephalotrichum*, *Cladosporium, Marquandomyces*, *Pseudogymnoascus*, *Purpureocillium*, and *Scopulariopsis* was based on the examination of cultures on DG18, MEA, and MLA incubated at RT for 7 d. All media contained 1.5% agar. Isolates that could be discriminated as separate taxa within genera but not identified to species were assigned numerical designations. Pycnidial and non-sporulating isolates were labeled “pycnidia” and “sterile”, respectively. Sporulating isolates that were not identified to the genus level were designated as “undetermined” (see Appendix A). Non-filamentous fungi and Mucoromycota, which were recovered infrequently on all media, were disregarded (see Duff et al. [10]).

Total nucleic acids were extracted from isolates of *Aspergillus*, *Penicillium*, *Pseudogymnoascus*, and *Talaromyces* using a FastDNA Spin Kit for Soil (MP Biomedicals, Irvine, CA, USA) following the manufacturer’s instructions. The region spanning the nuclear ribosomal internal transcribed spacer region (nuc rDNA ITS1-5.8S-ITS2 or ITS), 1100–1300 bp of the 3′ end of the nuclear ribosomal 28S rRNA gene (nuc 28S), and portions of the genes encoding for the proteins calmodulin (*CaM*) and ß-tubulin (*TUB*) were amplified using illustra PuReTaq Ready-To-Go PCR Beads (GE Healthcare, Little Chalfont, UK) following the manufacturer’s instructions. Amplification primers included (i) ITS4 and ITS5 [38] for the ITS, (ii) LR3F, LR3R, and LR7 [39,40] for the nuc 28S, (iii) Bt2a and Bt2b [41] for the *TUB* gene, and (iv) CMD5 and CMD6 [42] for the *CaM* gene. PCR products were purified using the EZNA Cycle Pure Kit (Omega Biotek, Norcross, Georgia) and quantified using a QuantiFluor dsDNA System (Promega, Madison, WI, USA) following the directions of the manufacturers. Sequencing reactions were performed using a BigDye terminator version 3.1 Cycle Sequencing Kit (Life Technologies, Austin, TX, USA) and the primers listed above.

Sequences were edited in Sequencher 4.7 (Gene Codes, Ann Arbor, Michigan) and compared to ITS sequences in GenBank [43] using BLAST (Basic Local Alignment Search Tool) [44]. The names adopted for taxa (see Appendix A) were based on matches to ITS sequences with ≥98% sequence similarity. Secondary barcodes (*TUB* and *CaM* sequences) were obtained to verify the identification of isolates of *Aspergillus europaeus*, *A. fructus*, *A. tubingensis*, *Penicillium sanguifluum*, *P. scabrosum*, and *P. yarmokense*.

### 2.3. Assessment of Xerotolerance

Xerotolerances of each sequence isolate of *Aspergillus*, *Penicillium*, *Pseudogymnoascus*, and *Talaromyces* (listed in Appendix A) were assessed based on their growth on malt yeast agar (MYA) [45] containing 1.5% agar and increasing molar concentrations of glycerol (MYA + G) or sucrose (MYA + S). We also assessed the xerotolerances of up to two additional representatives of each species.

To prepare MYA + G (1.0 M, 2.0 M, 3.0 M, 4.0 M, 5.0 M, 6.0 M, 7.0 M), the required amount of glycerol (*w*/*v*) was added to a 1 L volumetric flask. Dry ingredients for MYA, excluding agar, were dissolved in 200 mL of distilled water in a separate beaker, added to the volumetric flask, and mixed with the glycerol. After the final volume was adjusted to 1 L with distilled water, the contents of the flask were mixed and divided equally between two 1 L beakers. The pH of each solution was determined to ensure that they were between 6.8 and 7.0 prior to the addition of agar. MYA + S (0.5 M, 1.0 M, 1.5 M, 2.0 M, 2.5 M, 3.0 M) was prepared in a similar manner, except that the sucrose was dissolved via heating in distilled water and then cooled prior to its addition to a 1 L volumetric flask. Both media were autoclaved, cooled to 50 °C in a water bath, and poured into sterile, 90 mm plastic Petri dishes. Plates of osmoticant-augmented MYA were grouped according to their molar concentrations of glycerol or sucrose, conditioned for 24 h in a laminar flow hood, and inoculated within another 24 h. The a_w_ of MYA + G and MYA + S at each molar concentration was measured from two uninoculated 72-h-old plates and one uninoculated 7-day-old plate (see Appendix A) using a WP4C Water Potential Meter.

Conidial suspensions of isolates of *Aspergillus*, *Penicillium*, and *Talaromyces* were prepared by transferring 2 × 2 mm blocks of mycelium from the edges of sporulating cultures on MLA to 10 mL of a sterile solution containing 0.05% agar, 0.9% NaCl, and 0.05% Tween 20. The conidia of *Pseudogymnoascus pannorum* were suspended in the molten slurry produced by cooling a flame-sterilized transfer tool in the medium near the edges of growing colonies on MLA. Plates of MYA + G and MYA + S were point-inoculated centrally and in triplicate using these suspensions, stacked in Petri dish bags with plates of the same molar concentration of each osmoticant to maintain consistent a_w_, and incubated for 2 wk at 25 °C. The diameter of the reverse of each colony was measured every second day for 2 wk and used to calculate mean radial growth rates (mm d^−1^) [46]. Standard error was calculated for all means, but only values of 0.1 mm or greater are depicted in graphs. Goodness of fit was assessed using the coefficient of determination (*R^2^*).

### 2.4. Statistical Analyses

The car package version 3.0-11 [47] in R version 4.0.3 [48] was used to evaluate simple linear regression models for chemical and physio-chemical soil parameters, including a_w_, water content, pH, total carbon, total inorganic carbon, available nitrate, available phosphate, available sulfate, available potassium, and electrical conductivity. Significant outcomes were subjected to post hoc tests for multiple pairwise comparisons using the emmeans package version 1.6.2-1 [49]. The same package was used to adjust *p*-values according to the Benjamini–Hochberg (B-H) false discovery rate.

Colony-forming units per gram of soil (CFU g^−1^) and the relative abundance of each species were calculated from the numbers of fungal colonies recovered on DRBA, DRBAG, and MEA20S. Values for relative abundance were utilized to generate stacked bar plots and pie charts using the ggplot2 package version 3.3.5 [50]. Color scales in the viridis package version 0.5.1 [51] were applied to improve the readability of graphics for individuals with common forms of color vision deficiency.

Diversity indices were calculated using the vegan package version 2.5-7 [52]. We employed the car package for multivariate linear models to test the significance of the relationships between the dependent variables (species richness and species diversity), sampling sites (soils from the tops of mounds: M1, M2, M3; soil from within mounds: L1, L2, L3; soils from non-mounds: N1, N2, N3), and isolation media (DRBA, DRBAG, MEA20S). Subsequent post hoc pairwise comparisons were performed for significant outcomes for all combinations of sites using the emmeans package. *p*-values were adjusted according to the B-H false discovery rate employing the same package.

Morisita–Horn similarity indices were calculated employing the vegan package. These data were converted to distance matrices and used to generate dendrograms in R. A simple linear regression model in Microsoft Excel 365 version 16.0 was used to determine if there were significant differences in the number of Aspergillaceae isolated on different media. We then performed post hoc Tukey’s honest significant difference tests (HSD) in Excel for all pairwise comparisons of these data to identify significant differences (*p* < 0.05).

A permutational multivariate analysis of variance (PERMANOVA) in R was used to test for statistically significant differences in the composition of fungal communities from the different sampling sites (M, L, N). *p*-values for all pairwise comparisons were adjusted employing the B-H false discovery rate in the emmeans package, and permutational multivariate analyses of dispersion were run to confirm the observed values of significance.

## 3. Results

### 3.1. Soil Water Activity, Water Content, pH, and Nutrient Analyses

The water activity and water content of soils from the tops of nests (M) and within the nests (L) of *Formica obscuripes* were lower than from non-mound (N) sites (Table 1), whereas electrical conductivities and the levels of total carbon, total organic carbon, and available nitrate, phosphorus, potassium, and sulfate were higher (Table 2). Soils from both the tops of mound nests (M) and within mounds (L) differed significantly (*p* < 0.05) from those of non-mound sites (N) with respect to all measured parameters except pH, available nitrate, and electrical conductivity.

Differences in the sizes of the particles of soils from the tops of mounds (M) and non-mound sites (N) were not significant (Table 3), and there were insufficient quantities of soil from within mounds (L) to perform particle size analysis.

### 3.2. Composition, Richness, and Diversity of Fungal Communities

A total of 3594 isolates, including 98 species representing 19 families of Ascomycota, were recovered on DRBA (a_w_ = 0.998), DRBAG (a_w_ = 0.944), and MEA20S (a_w_ = 0.987) from the soils of mound nests (M, L) and non-mound (N) sites (Table 4). The highest number of isolates (1877) was recovered on DRBAG, and most of these (1263) were obtained from soils from the uppermost 3 cm of mounds (M1, M2, M3).

The structure of the fungal communities between sites differed significantly (PERMANOVA; *p* < 0.01) (Appendix A), and species assemblages from mounds (M, L) were more similar to each other than to non-mound (N) sites, with the exception of one community from mound tops (M3) isolated on MEA20S (Figure 1, Appendix A). Morisita–Horn similarity indices for the fungal communities from the tops of mounds (M), within mounds (L), and non-mound (N) soils differed significantly (*p* = 0.0002), as did the communities isolated on DRBA, DRBAG, and MEA20S (PERMANOVA; *p* = 0.022), but we detected no significant interaction between site and isolation medium.

The communities of fungi from mound tops (M) and within mounds (L) on DRBA, DRBAG, and MEA20S were dominated by Aspergillaceae (species of *Aspergillus* and *Penicillium*) (Figure 2, Table 5), a family that accounted for 12.9 (M3 on MEA20S) to 100% (L2 on DRBAG) of the strains recovered from these two sites (Appendix A). The numbers of Aspergillaceae isolated on DRBA, DRBAG, and MEA20S varied among sites (*p* = 0.027), but the post hoc Tukey’s HSD test indicated that only DRBAG and MEA20S yielded significantly different numbers of the members of this family. The greatest numbers of Aspergillaceae were isolated on DRBAG, with the highest percentages of the representatives of this family (51.3–100%) found in soils from nests (M, L). This relationship is illustrated in the dendrogram of fungi from different families recovered on DRBAG (Figure 1), which is based on nearly three times the number of Aspergillaceae isolated on DRBA or MEA20S (Appendix A).

Species of *Penicillium* were the most abundant Aspergillaceae recovered on all isolation media (Appendix A), but a greater number of representatives of this genus were isolated from non-mound (N) soils than from mounds (M, L) on DRBAG and MEA20S. Of the fourteen members of this genus identified to species, the most common were *P. citrinum* (5.1–56.8%) and *P. pasqualense* (1.7–12.9%). *Penicillium citrinum* was recovered from all sites using each isolation medium and was most abundant in soils from mounds (M, L) (i.e., up to 50% on DRBAG and 56.8% on MEA20S).

While less abundant than species of *Penicillium*, members of the genus *Aspergillus* were recovered consistently and in high numbers from the tops of nests (M) and from within mounds (L) on all media. The most abundant species, *A. fructus*, was recovered from all sites on DRBAG and represented 25.6% of isolates from within nests (L) on this medium (Appendix A). *Aspergillus insuetus* and *A. tubingensis* were also abundant in soils from mounds (M, L), with the latter species representing 43.6% of isolates recovered from the tops of mounds (M) on DRBAG. The least abundant member of the genus, *A. europaeus*, was isolated only on DRBA and represented less than 5.6% of isolates recovered on this medium.

Trichocomaceae, represented by *Talaromyces atricola* and *T. neorugulosus*, were less abundant than Aspergillaceae (Appendix A). These species were recovered from soils from mounds (M, L) and non-mound (L) sites (Appendix A), but they never comprised more than 3.4% of isolates recovered on DRBA, DRBAG, or MEA20S. *Pseudogymnoascus pannorum* (Pseudeurotiaceae) was present in all soils from mounds (M, L) and non-mound (L) sites, but was isolated most frequently from within mounds (L), where it accounted for up to 34.1% of recovered strains on DRBA. Most of the isolates of this species conformed to the descriptions provided by Domsch et al. [53] and Samson et al. [15], but we distinguished a smaller number of strains as “*Ps. pannorum* strain 2” based on their more lightly pigmented colonies on MLA (not illustrated).

Other taxa comprising more than 10% of isolates from any individual sampling site included Chaetothyriales (*Exophiala*), Cladosporiales (*Cladosporium*), Hypocreales (*Acremonium*-like taxa, *Albifimbria*, *Clonostachys*, *Cylindrocarpon*, fusarium-like taxa, *Gliocladium*, *Marquandomyces*, *Purpureocillium*, *Tolypocladium*, *Trichoderma*, *Volutella*), and Microascales (*Doratomyces*, *Microascus*, *Scopulariopsis*, *Wardomyces*) (Appendix A). Cladosporiales were most abundant in soils from mound tops (M) (up to 56.4%) and non-mound (N) sites (up to 11.1%) on MEA20S, while Hypocreales comprised 11.1–52.6% of strains recovered from non-mounds (N) on all isolation media. The majority of isolates belonging to the latter order (*Marquandomyces marquandii* and *Purpureocillium lilacinum*) were isolated on DRBAG and MEA20S. *Exophiala* represented 9.1–13.3% of isolates in non-mound (N) soils on DRBA, but this genus was less frequently isolated or absent from other sites on DRBAG and MEA20S. Sterile and undetermined fungi were isolated most frequently on DRBA and were less abundant or absent on all media in soils from within mounds (L).

Fungal species richness and diversity were highest in soils from the tops of nests (M) on DRBA and from non-mound (N) sites on DRBAG, respectively, but both measures varied between sites (Table 6). Richness differed significantly between all sites on DRBA (*p* < 0.05), between inner mound (L) and non-mound (N) soils on DRBAG (*p* < 0.01), and between all sites except the tops of mounds (M) and non-mound (N) sites on MEA20S (*p* < 0.01). There were also significant differences in species diversity, as measured by the Shannon index, between mound (M, L) and non-mound (N) soils on DRBAG (*p* < 0.05), and between inner mound (L) and non-mound (N) soils on MEA20S (*p* = 0.027). Simpson diversity and Simpson inverse diversity indices between sites did not differ significantly. Multiple linear regression analyses indicated significant differences in species richness between sites (*p* = 0.001), between fungal communities isolated on different media (*p* = 0.045), and in the interaction between sites and media with respect to species richness (*p* = 0.001) and diversity (*p* = 0.014).

### 3.3. Assessment of Xerotolerance

Every isolate tested for xerotolerance (listed in Appendix A) grew on unaugmented MYA (a_w_ = 1.00) and had mean radial growth rates greater than 1.0 mm d^−1^ on this medium, except *Penicillium charlesii*, *P. parvulum*, *Pseudogymnoascus pannorum*, *Talaromyces atricola*, and *T. neorugulosus* (Figure 3). Mean growth rates of all taxa were higher on MYA containing an osmoticant, and most species grew more rapidly on MYA + 1 M sucrose (a_w_ = 0.97) than on MYA + 1 M glycerol (a_w_ = 0.97) (Appendix A).

All members of the Aspergillaceae included in our tests were xerophilic (i.e., capable of growth on media at or below a water activity of 0.85) and most of the species we assessed exhibited faster radial growth on MYA supplemented with sucrose (Figure 3, Appendix A). The highest rates of growth were exhibited by *Aspergillus tubingensis* (7.06 mm d^−1^) and *Penicillium thomii* (4.73 mm d^−1^) on MYA + 0.5 M sucrose (a_w_ = 0.99), but species of *Aspergillus* generally grew faster than members of the genus *Penicillium* on MYA amended with the highest tolerated concentration of sucrose (MYA + 2.5 M sucrose, a_w_ = 0.78). *Talaromyces atricola* and *T. neorugulosus* grew more slowly than Aspergillaceae on glycerol- and sucrose-augmented MYA; both species exhibited limited growth on sucrose-augmented MYA, with a water activity of 0.89, but failed to grow on MYA containing 2.5 M sucrose (a_w_ = 0.78) or 5 M glycerol (a_w_ = 0.84).

*Pseudogymnoascus pannorum* exhibited the slowest rates of growth on MYA and osmoticant-augmented MYA and was the least xerotolerant species assessed. None of the strains of *Ps. pannorum* grew on MYA containing 4.0 M glycerol (a_w_ = 0.89), but isolates of *Ps. pannorum* strain 2 differed from those identified as *Ps. pannorum* strain 1 in their ability to grow on sucrose-augmented MYA, with a water activity of 0.90. These isolates also exhibited less variable growth responses on MYA and MYA containing up to 1 M of osmoticant (a_w_ = 1.00 − 0.97) and had faster rates of radial growth (0.71 mm d^−1^ versus 0.36 mm d^−1^ on MYA and 1.04 mm d^−1^ versus 0.43 mm d^−1^ on MYA + 0.5 M sucrose).

## 4. Discussion

### 4.1. The Abundance and Diversity of Fungi in Soils from Mound Nests

The results of this investigation agree with previous studies that recovered higher numbers of fungi from the nests of mound-building species of *Formica* than non-nest sites and reported significant differences in the composition of fungal communities from mound nests and non-nest soils [10,12,54]. We also observed significant differences between the assemblages of fungi recovered on reduced-water-activity media (a_w_ = 0.944, a_w_ = 0.987) and that fungal species richness and diversity varied between soils from the tops of mounds (M), soil from within mounds (L), and soils from non-mounds (N).

The present study also documents significant differences in the water characteristics and chemical properties of soils from the mound nests of *F. obscuripes* soils from non-mounds. Soils from the tops of mounds and within mounds had lower water activities (a^w^ ≤ 0.82), contained less water as measured by percentage of water content, and were more acidic than non-mound soils. Nest soils also had higher levels of nutrients and electrical conductivities. These results agree with previous reports of ant-mediated changes to soil characteristics, particularly nutrient enrichment, that are attributed largely to the accumulation of organic material by mound-building and soil-dwelling ants [3,4,8,9,55,56,57,58].

Our discovery that the mound nests of *F. obscuripes* are reduced-water-activity environments supports the suggestion [10] that water availability is an important factor influencing the structure of fungal communities in the nests of this and possibly other species of *Formica*. This conclusion is reinforced by the recovery of higher numbers of xerophilic Aspergillaceae, a family of Ascomycota predominant in reduced-water-activity and xeric environments [20,22], from mound nests. In our investigation, species of *Aspergillus* and *Penicillium* comprised 29.1–100% of isolates recovered from soils from mound nests (M1, M2, M3, L1, L2, L3) on DRBA (a_w_ = 0.998) and on media with reduced water activity (DRBAG a_w_ = 0.944, MEA20S a_w_ = 0.987).

The genus *Aspergillus* encompasses the most xerotolerant Aspergillaceae, and includes species isolated frequently from alpine and desert soils, the built environment, and a variety of low-water-activity substrates (e.g., dried fruit, stored grain, sugar-preserved foods) [14,15,16,19,20,22]. Species of *Aspergillus* are also abundant in the nests of soil-inhabiting and mound-building ants [10,11], and in the current study accounted for up to 50% of the fungi recovered from mound nests on DRBAG. All of the members of this genus assessed for xerotolerance in our investigation were capable of growing on MYA with 2.5 M S (a_w_ = 0.78) and 5.0 M G (a_w_ = 0.84). *Aspergillus fructus* (*A. versicolor fide* Sklenář et al. [59]), the most abundant member of the genus from the nest mounds of *F. obscuripes*, is a xerophile documented from low-humidity environments, clinical specimens, fruit, house dust, and soil [19,59,60,61,62], but it has not been reported previously from animal-modified habitats. *Aspergillus fructus* was present in lower numbers in non-mound soils, while *A. europaeus* and *A. insuetus*, two species found commonly in soils [63,64], were isolated from the nest soils of *F. obscuripes* but never recovered from non-mound sites. These results mirror those of Duff et al. [10], who reported that species of *Aspergillus* accounted for 42.4–80.4% of isolates from the mound nests of *F. ulkei* but were absent or represented only 0.3% of strains from non-nest locations.

Species of *Penicillium* accounted for up to 66.7% of isolates from non-mound sites and were more abundant in soils from non-mound sites than soils from mounds on DRBAG and MEA20S. Only two species, *P. sizovae* and *P. thomii,* were restricted to soils from mounds. Duff et al. [10] also recovered more species of *Penicillium* than *Aspergillus* and reported that isolates of the former genus were more abundant in non-mound soils. Penicillia are common in soils, foods, and the built environment, and many species tolerate low-water activities [15,53]. In the current study, every member of the genus assessed for xerotolerance was capable of growth on MYA with 2.5 M S (a_w_ = 0.78), and all but two species (*P. estinogenum* and *P. skrjabinii*) grew on MYA of 5.0 M G (a_w_ = 0.84). Although the xerotolerances of many of the species of *Penicillium* we isolated have not been reported previously, our results support the water-activity limits described for *P. chrysogenum sensu lato* and *P. citrinum* [22,65,66]. The former species has been recovered in high numbers from hypersaline environments [15,67], whereas the latter is reported from cereals, dried foods, decaying plant material, and soil [22,53]. The 12 species of *Penicillium* we recovered from both mound and non-mound soils are not characteristic of reduced-water-activity environments.

*Pseudogymnoascus pannorum* was abundant in both mound and non-mound soils. This slow-growing, psychrotolerant species is common in temperate, arctic, and alpine soils [16,53,68] and exhibits considerable morphological and physiological variation [52,69]. We recovered *Ps. pannorum* from all sites, but the largest numbers of this species were isolated from nest mounds on DRBA. Our finding that most of these isolates resembled the strain we determined to be incapable of growth on media with a water activity at or below a_w_ = 0.89 (MYA + G) or a_w_ = 0.90 (MYA + S) (*Ps. pannorum* strain 1) supports the observation that *Ps. pannorum* is not xerophilic [16,65].

*Talaromyces atricola* and *T. neorugulosus* were the least abundant of all of the fungi we assessed for xerotolerance. Both taxa were restricted largely to soils from the mound nests of *F. obscuripes* and grew slowly on osmoticant-augmented media (i.e., ≤0.3 mm d^−1^ on MYA + S, a_w_ = 0.90), supporting the observation that species of *Talaromyces* are generally less xerophilic than members of the Aspergillaceae [22,66,70].

Fungal taxa recovered in large numbers that we did not assess for xerotolerance included members of the orders Cladosporiales and Hypocreales. Xerophily is documented in several species of *Cladosporium* [22,70], and representatives belonging to the complexes *Cladosporium cladosporioides*, *C. herbarum*, and *C. sphaerospermum* accounted for 3.5–50.1% of the isolates recovered from soils from the tops of individual mounds on MEA20S. These and other *Cladosporium* were less abundant or absent in soils from within mounds and non-mound sites. In contrast, members of the Hypocreales were more abundant in non-mound soils, where they represented 11.1–52.6% of strains recovered from individual sites. *Marquandomyces marquandii* and *Purpureocillium lilacinum* were the most frequently isolated members of this order in non-nest soils. Xerotolerance has been reported for both species [71,72], so it is unsurprising that the highest numbers of these taxa were recovered on DRBAG and MEA20S.

### 4.2. Water Activity Is an Important but Poorly Explored Determinant of the Structure of Fungal Communities

Water content, or the mass fraction of water present in materials, has long been recognized as critical in governing the growth of soil-inhabiting microorganisms (for example, see Sewell [73]). Gravimetric water content is determined by the comparison of sample mass before and after drying to constant weight, but this measure is not predictive of the ability of microorganisms to grow when it is used alone and without knowledge of the sorption characteristics of materials and the solute composition of the water contained therein. This is because the solute composition of contained water, particularly the electrochemical characteristics of constituent solutes, determines the osmotic properties of materials and the availability of water that is free to support extracellular microbial processes [74]. The quantity of free water available in a substrate, including adsorbed water held loosely in spaces by capillary action or bound weakly by molecular attraction to other materials, is referred to as water activity [21], and it is directly proportional to equilibrium relative humidity (ERH) measured proximate to the surface [75,76].

The value of water activity for predicting microbial growth was first appreciated by food scientists who understood the importance of water availability in food preservation [72,77] and the close correlation of this measure with certain structural characteristics of food (e.g., crunch and crispness) [78]. Water availability, expressed as a_w_ or ERH, is also preferred to other measures that describe the biological potential of water in the domains of building science and the analyses of the growth of microorganisms in indoor environments. Building materials vary in their chemical composition, physical complexity, and adsorptive and water-holding capacities [79]. It has also long been recognized that materials with the same water content can possess different water activity values, just as materials with the same water activity values can differ in their water contents [21,70,74,76,80]. Furthermore, because of the hysteretic nature of the sorption curves of many porous substrates, dry materials that take up water will reach water activities that may differ from the values they attain after drying to the same water content following flooding. Finally, because the distribution of water in building materials and other structurally complex substrates is normally not uniform, materials subject to different types of wetting or with different water activities will support the growth of cohorts of organisms with differing requirements for available water [21,23].

Our investigation of the composition of the mycota of the mound nests of *Formica obscuripes* supports the use of both standard or higher-water-activity media and media with reduced water activity to recover fungi from natural environments. Media used commonly in the enumeration of microorganisms from environmental samples and foods have water activities higher than 0.98 [81,82], but the recovery of xerophilic fungi from these substrates is enhanced greatly using media of only incrementally reduced water activity (i.e., a_w_ = 0.94 − 0.95) [17,66,83,84,85]. This is because the growth optima of both hydrophilic and many xerophilic fungi are above a_w_ 0.90, even though members of the latter group have lower a_w_ minima [21]. Given the heterogenous composition of most natural substrates and the expectation that the distribution of water within them is spatially and temporally non-uniform, it is reasonable to assume that xerophilic fungi capable of growing over a range of water activities would be better adapted to reduced water environments than hydrophilic taxa possessing higher a_w_ minima.

The nests of mound-building ants and other natural, animal-modified habitats are unexplored reservoirs of fungal diversity that warrant further study using high- and reduced-water-activity isolation media. A fuller picture of this diversity would be obtained by identifying all the fungi isolated from these habitats and by using sequence-based approaches to identify micromorphologically similar members of species complexes, non-sporulating and non-culturable taxa, and species that are traditionally under sampled using culture-dependent methods. Finally, a better understanding of the distribution of fungi in natural environments could be achieved by determining the water activities of the substrates from which they are recovered and by employing solute-augmented growth media to assess the xerotolerances of recovered isolates.

## Figures and Tables

**Figure 1 jof-10-00735-f001:**
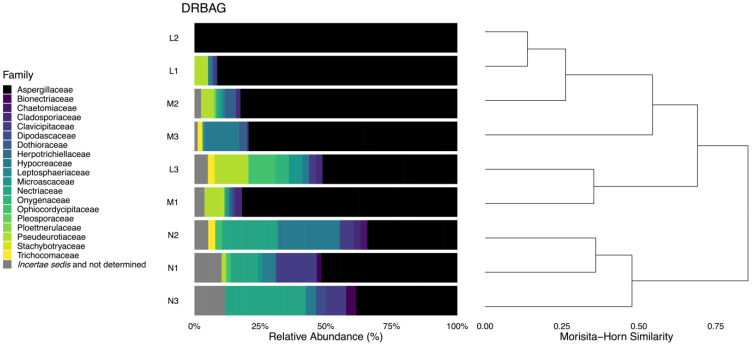
Relative abundance of representatives of families of fungi (bar graphs) and hierarchical clustering (dendrogram) of fungal communities from different soils (M = tops of mound nests, L = within nests, N = non-nest mound sites) isolated on DRBAG.

**Figure 2 jof-10-00735-f002:**
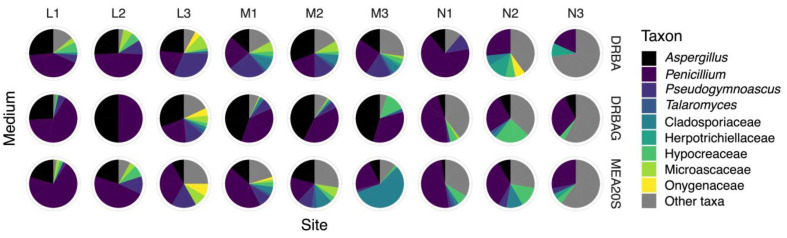
Pie charts illustrating the relative abundances of 4 genera and 5 families of fungi from different soils (M = tops of mound nests, L = within nests, N = non-nest mound sites) on three isolation media (DRBA, DRBAG, MEA20S).

**Figure 3 jof-10-00735-f003:**
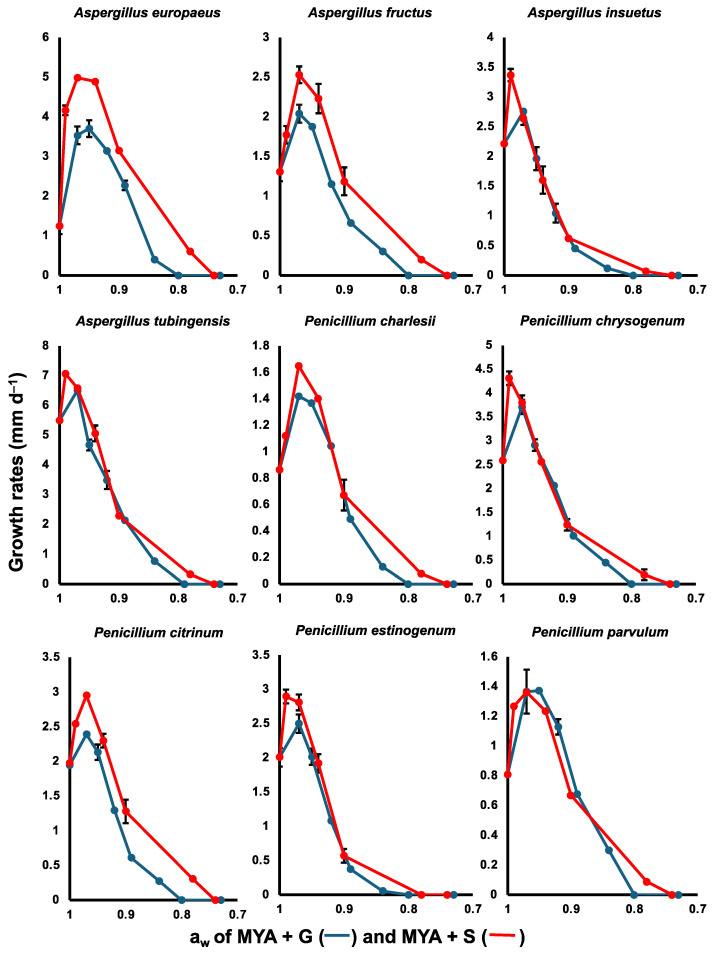
Mean growth rates (mm d^−1^ ± SE) of species of *Aspergillus*, *Penicillium*, *Pseudogymnoascus*, and *Talaromyces* on MYA with different water activities (a_w_) resulting from the addition of glycerol (blue lines) and sucrose (red lines).

**Table 1 jof-10-00735-t001:** Water activity (a_w_), water content (WC), and pH of soils (mean ± SD) from the tops of mound nests (M), within nests (L), and non-mound (N) sites.

Property	M (*n* = 3)	L (*n* = 3)	N (*n* = 3)	Site Effect (d.f. = 2, 6)
a_w_	0.75 ± 0.11 ^N^	0.82 ± 0.03 ^N^	1.00 ± 0.00 ^M,L^	11.79 **
WC (%)	10.93 ± 0.45 ^N^	13.03 ± 1.46 ^N^	19.20 ± 1.67 ^M,L^	32.47 ***
pH	6.33 ± 0.12 ^L,N^	5.89 ± 0.13 ^M,N^	6.86 ± 0.04 ^M,L^	63.85 ***

Superscript text indicates sites differing significantly in post hoc comparisons of soils; ***, *p* < 0.001; **, *p* < 0.01.

**Table 2 jof-10-00735-t002:** Chemical properties (mean ± SD) of soils from the tops of mound nests (M), within nests (L), and non-mound (N) sites.

Property	M (*n* = 3)	L (*n* = 3)	N (*n* = 3)	Site Effect (d.f. = 2, 6)
Total carbon (%)	15.23 ± 2.35 ^N^	11.57 ± 1.53 ^N^	5.71 ± 0.47 ^M,L^	25.77 **
Total organic carbon (%)	14.20 ± 2.50 ^N^	10.67 ± 1.26 ^N^	3.98 ± 0.18 ^M,L^	30.80 ***
Available nitrate (mg kg^−1^)	88.23 ± 9.61 ^ns^	163.67 ± 91.08 ^N^	4.77 ± 1.70 ^L^	6.78 *
Available phosphorus (mg kg^−1^)	91.33 ± 30.29 ^N^	124.67 ± 30.37 ^N^	3.53 ± 1.07 ^M,L^	19.14 **
Available potassium (mg kg^−1^)	1173.33 ± 395.77 ^N^	1620.00 ± 355.95 ^N^	339.33 ± 22.30 ^M,L^	13.40 **
Available sulfate (mg kg^−1^)	27.87 ± 6.63 ^N^	37.00 ± 8.54 ^N^	6.07 ± 0.49 ^M,L^	19.40 **
Electrical conductivity (ds m^−1^)	0.71 ± 0.16 ^L^	1.74 ± 0.55 ^M,N^	0.32 ± 0.01 ^M^	14.60 **

Superscript text indicates sites differing significantly in post hoc comparisons of soils; ***, *p* < 0.001; **, *p* < 0.01; *, *p* < 0.05, ^ns^, no significance.

**Table 3 jof-10-00735-t003:** Particle size comparison of soils from the tops of mound nests (M) and non-mound (N) sites. There were insufficient quantities of soil from within mounds (L) to perform particle size analysis.

Particle Size	M (*n* = 3)	N (*n* = 3)
% Clay (<4 µm)	22.6 ± 3.8	24.5 ± 2.0
% Silt (4 µm–0.063 mm)	55.5 ± 2.4	49.6 ± 5.4
% Fine sand (0.063–0.2 mm)	9.2 ± 1.4	8.5 ± 0.4
% Coarse sand (0.2–2.0 mm)	12.6 ± 3.1	15.1 ± 2.6

**Table 4 jof-10-00735-t004:** Numbers of colonies recovered on DRBA, DRBAG, and MEA20S from soils from the tops of mound nests (M), within nests (L), and non-mound (N) sites.

Medium	M1	M2	M3	L1	L2	L3	N1	N2	N3	Total
DRBA (a_w_ = 0.998)	139	116	139	168	91	185	22	30	16	906
DRBAG (a_w_ = 0.944)	343	391	529	179	13	124	158	91	49	1877
MEA20S (a_w_ = 0.987)	147	76	156	89	51	25	111	98	58	811
Total	629	583	824	436	155	334	291	219	123	3594

**Table 5 jof-10-00735-t005:** Numbers of isolates belonging to the Aspergillaceae recovered on DRBA, DRBAG, and MEA20S from soils from the tops of mound nests (M), within nests (L), and non-mound sites (N).

Medium	M1	M2	M3	L1	L2	L3	N1	N2	N3	Total
DRBA (a_w_ = 0.998)	46	69	65	127	71	80	20	5	2	485
DRBAG (a_w_ = 0.944)	287	341	430	167	13	67	91	39	19	1454
MEA20S (a_w_ = 0.987)	97	36	46	81	34	8	63	50	17	432
Total	430	446	541	375	118	155	174	94	38	2371

**Table 6 jof-10-00735-t006:** Species richness and diversity measures (mean ± SE) of communities of fungi isolated from soils from the tops of mound nests (M), within nests (L), and non-mound (N) sites.

Site	Medium	Species Richness	Shannon Diversity	Simpson Diversity	Simpson Inverse
M	DRBA	27.33 ± 3.48 ^L,N^	2.97 ± 0.10 ^ns^	0.93 ± 0.01 ^ns^	13.79 ± 0.90 ^ns^
	DRBAG	18.67 ± 1.45 ^ns^	2.05 ± 0.23 ^N^	0.81 ± 0.05 ^ns^	5.96 ± 1.46 ^ns^
	MEA20S	22.33 ± 2.40 ^L^	2.65 ± 0.28 ^ns^	0.84 ± 0.07 ^ns^	9.49 ± 4.02 ^ns^
L	DRBA	19.67 ± 2.60 ^M,N^	2.43 ± 0.04 ^ns^	0.86 ± 0.02 ^ns^	7.13 ± 0.87 ^ns^
	DRBAG	11.67 ± 4.48 ^N^	1.83 ± 0.43 ^N^	0.76 ± 0.07 ^ns^	5.17 ± 1.77 ^ns^
	MEA20S	10.67 ± 0.88 ^M, N^	1.96 ± 0.22 ^N^	0.79 ± 0.08 ^ns^	6.10 ± 1.67 ^ns^
N	DRBA	10.33 ± 2.33 ^M,L^	2.20 ± 0.31 ^ns^	0.86 ± 0.06 ^ns^	9.36 ± 2.83 ^ns^
	DRBAG	25.33 ± 1.67 ^L^	2.99 ± 0.09 ^M,L^	0.93 ± 0.01 ^ns^	15.82 ± 2.83 ^ns^
	MEA20S	23.33 ± 1.76 ^L^	2.93 ± 0.06 ^L^	0.93 ± 0.01 ^ns^	14.22 ± 1.21 ^ns^

Superscript text indicates sites differing significantly (*p* < 0.05) in post hoc comparisons of soils; ^ns^, no significance.

## Data Availability

All data of this research are reported in the article and Appendix A.

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
