# Peer review of "Xerophilic Aspergillaceae Dominate the Communities of Culturable Fungi in the Mound Nests of the Western Thatching Ant (Formica obscuripes)"

_jof, 2024, doi:10.3390/jof10110735_

Round 1

Reviewer 1 Report

This manuscript studies the abundance and diversity of fungi in soils from mound nests and non-mound, and combines soil water activity, water content, pH and nutrient analysis, as well as isolation and further analysis of fungi, and determines that Soils from mound nests are dominated by members of the Aspergillaceae. and believes that water availability is an important factor influencing the structure of fungal communities in these animals-modified habitats. The study is well-conceived and provide important insights into the interactions between fungi and water in animal habitats. The research is extensive and practical, and these findings have a significant impact on unexplored reservoirs of fungal diversity.

Specific Comments:

Abstract

1.The Abstract section does not provide background.It would be better that background is added.

Introduction

2.From line 67 to 79,this part of the presentation of the research could be more concise.

3.Lack of introduction to the effect of water on fungi.

Materials and Methods

4.The part of statistical analysis could be dispersed into corresponding experiments rather than listing them separately.

Result:

5.Figure 1.If it is possible, The font in the picture should be made bigger for the readers' convenience.

6.Figure 2.The picture is not clear.The picture may be better arranged vertically.

7.Figure 3. Some Line graphs have error bars and some Line graphs do not.All line charts should have error bars.

Discussion

8. The discussion section can be expanded to include the relationship between plants and microorganisms for more in-depth analysis and interpretation of the results.

Conclusion

9.The manuscript does not provide conclusions and any direction on future research or the next steps following this study's completion. Please consider providing some guidance in this area.

Author Response

Reviewer 1

 I would not like to sign my review report

 I would like to sign my review report

Does the title describe the article's topic with sufficient precision?

Yes

No

Does the introduction provide a comprehensive yet concise overview about the state of knowledge in the area of research?

Yes

No

Is the research design appropriate and are the methods adequately described?

Yes

No

Are the results presented clearly and in sufficient detail, are the conclusions supported by the results and are they put into context within the existing literature?

Yes

No

Are all of the cited references relevant to the research?

Yes

No

Does this article provide a relevant contribution to the scientific discussion of this topic?

Yes

No

English language and style

I am not qualified to assess the quality of English in this paper.

The English is very difficult to understand/incomprehensible.

Extensive editing of English language required.

Moderate editing of English language required.

Minor editing of English language required.

English language fine. No issues detected.

Comments for Authors

Advice for completing your review can be found at: https://www.mdpi.com/reviewers#Review_Report

Major comments

This manuscript studies the abundance and diversity of fungi in soils from mound nests and non-mound, and combines soil water activity, water content, pH and nutrient analysis, as well as isolation and further analysis of fungi, and determines that Soils from mound nests are dominated by members of the Aspergillaceae. and believes that water availability is an important factor influencing the structure of fungal communities in these animals-modified habitats. The study is well-conceived and provide important insights into the interactions between fungi and water in animal habitats. The research is extensive and practical, and these findings have a significant impact on unexplored reservoirs of fungal diversity.

Detail comments

Specific Comments:

Abstract

1.The Abstract section does not provide background. It would be better that background is added. - Done. The key words have been changed to reflect this emendation.

Introduction

2.From line 67 to 79, this part of the presentation of the research could be more concise. - We think it’s important to specify the questions we attempted to answer in this study.  And, given the varied applications of terms used to describe water relationships in Fungi, we also think it’s important to define what we mean by describing a fungus as xerophilic.

3.Lack of introduction to the effect of water on fungi. - Although the effects of water on Fungi is of tremendous interest to the authors, we feel discussion of this topic lies outside the scope of our study.

 Materials and Methods

4.The part of statistical analysis could be dispersed into corresponding experiments rather than listing them separately. - We disagree and have elected to retain this section of the Materials and Methods as written originally.

 Results

5.Figure 1.If it is possible, The font in the picture should be made bigger for the readers' convenience. - The font has been darkened and enlarged.

6.Figure 2.The picture is not clear. The picture may be better arranged vertically. - The orientation of this figure has been adjusted.

7.Figure 3. Some Line graphs have error bars and some Line graphs do not. All line charts should have error bars. - Error bars represent the variability of data.  Where the measurable growth rates of the isolates assessed for xerotolerance were invariant, there are no error bars.

 Discussion

  1. The discussion section can be expanded to include the relationship between plants and microorganisms for more in-depth analysis and interpretation of the results. - Again, while this topic is significant and of great interest to us, it lies outside the scope of our study. It should also be noted that we did not document the vegetation in the areas where the sampled nests are located.

 Conclusion

9.The manuscript does not provide conclusions and any direction on future research or the next steps following this study's completion. Please consider providing some guidance in this area. - We elected not to use a Conclusion section because the Discussion is not unusually long or complex.  Directions for future research are outlined in the last paragraph of the Discussion.

Reviewer 2 Report

Review

of the article “Xerophilic Aspergillaceae Dominate the Communities of Fungi Recovered from the Mound Nests of the Western Thatching Ant (Formica obscuripes)”

by the authors Rachelle M. Gross, Courtney L. Geer, Jillian D. Perreaux, Amin Maharaj, Susan Du, James A. Scott and Wendy A. Untereiner

Microbial communities in ant mound nests and the influence of ants on soil mycobiota have not yet been sufficiently studied. The aim of the authors' research was to: (1) confirm that the community of culturable, filamentous Ascomycota from the mound nests of the western thatching ant (Formica obscuripes) differs from the assemblage of Fungi in adjacent, non-nest soils, (2) test the prediction that soils from nests have a lower water activity than those from non-mound sites, (3) evaluate if mound nests harbor greater numbers and a higher diversity of xerophilic Fungi, and (4) determine the xerotolerances of selected isolates of the most abundant taxa recovered from mound nests (i.e., species of Aspergillus, Penicillium, Pseudogymnoascus, and Talaromyces).

The authors obtained new data on the composition of fungal communities in the mound nests of the Western Thatching Ant (Formica obscuripes). Authors recovered 3594 isolates of filamentous Ascomycota from the mound nests of Formica obscuripes and adjacent, non-nest sites.

Soils from mound nests were dominated by members of the Aspergillaceae.

Fungi species found in ant nests were assessed for xerotolerance. For the first time, xerotolerance of some taxa was determined. The importance of free water for fungi is discussed. The authors' results are consistent with literature data that F. obscuripes nests are low-water environments, and support the suggestion that water availability is an important factor influencing the structure of fungal communities in these animal-modified habitats.

The “Abstract” outlines the main points of the presented work. The “Introduction” section provides a review of the literature, identifies the problem, and presents the objectives of the study.

The “Materials and Methods” section is written in sufficient detail and is understandable. Statistical processing of the obtained data was carried out. The following were used: car package version 3.0-11 (Fox and Weisberg 2018) in R version 4.0.3 (R Core Team 2020), emmeans package version 1.6.2-1 (Lenth 2021), ggplot2 package version 3.3.5 (Wickham 2016), vegan package version 2.5-7 (Oksanen et al. 2020).

The “Results” section is written in detail and in clear language. The presented results are illustrated with three figures and six tables. There are also Supplementary, which include six tables and two figures. The Discussion is based on the data obtained. The list of references includes 89 sources. All of them are related to the presented work.

Notes.

The manuscript does not have a separate "Conclusion" section. I suggest that the authors consider creating a separate "Conclusion" section where the main findings could be summarized. This could be the last paragraph of the "Discussion" section, separated into the "Conclusion" section.

After minor editorial corrections and additions, the manuscript may be published in the Journal of Fungi, special issue “Diversity and Ecology of Fungi from Underexplored and Extreme Environments”.

See above.

Author Response

Reviewer 2

 I would not like to sign my review report

 I would like to sign my review report

Does the title describe the article's topic with sufficient precision?

Yes

No

Does the introduction provide a comprehensive yet concise overview about the state of knowledge in the area of research?

Yes

No

Is the research design appropriate and are the methods adequately described?

Yes

No

Are the results presented clearly and in sufficient detail, are the conclusions supported by the results and are they put into context within the existing literature?

Yes

No

Are all of the cited references relevant to the research?

Yes

No

Does this article provide a relevant contribution to the scientific discussion of this topic?

Yes

No

English language and style

I am not qualified to assess the quality of English in this paper.

The English is very difficult to understand/incomprehensible.

Extensive editing of English language required.

Moderate editing of English language required.

Minor editing of English language required.

English language fine. No issues detected.

Comments for Authors

Advice for completing your review can be found at: https://www.mdpi.com/reviewers#Review_Report

Major comments

Microbial communities in ant mound nests and the influence of ants on soil mycobiota have not yet been sufficiently studied. The aim of the authors' research was to: (1) confirm that the community of culturable, filamentous Ascomycota from the mound nests of the western thatching ant (Formica obscuripes) differs from the assemblage of Fungi in adjacent, non-nest soils, (2) test the prediction that soils from nests have a lower water activity than those from non-mound sites, (3) evaluate if mound nests harbor greater numbers and a higher diversity of xerophilic Fungi, and (4) determine the xerotolerances of selected isolates of the most abundant taxa recovered from mound nests (i.e., species of Aspergillus, Penicillium, Pseudogymnoascus, and Talaromyces).

The authors obtained new data on the composition of fungal communities in the mound nests of the Western Thatching Ant (Formica obscuripes). Authors recovered 3594 isolates of filamentous Ascomycota from the mound nests of Formica obscuripes and adjacent, non-nest sites.

Soils from mound nests were dominated by members of the Aspergillaceae.

Fungi species found in ant nests were assessed for xerotolerance. For the first time, xerotolerance of some taxa was determined. The importance of free water for fungi is discussed. The authors' results are consistent with literature data that F. obscuripes nests are low-water environments and support the suggestion that water availability is an important factor influencing the structure of fungal communities in these animal-modified habitats.

The “Abstract” outlines the main points of the presented work. The “Introduction” section provides a review of the literature, identifies the problem, and presents the objectives of the study.

The “Materials and Methods” section is written in sufficient detail and is understandable. Statistical processing of the obtained data was carried out. The following were used: car package version 3.0-11 (Fox and Weisberg 2018) in R version 4.0.3 (R Core Team 2020), emmeans package version 1.6.2-1 (Lenth 2021), ggplot2 package version 3.3.5 (Wickham 2016), vegan package version 2.5-7 (Oksanen et al. 2020).

The “Results” section is written in detail and in clear language. The presented results are illustrated with three figures and six tables. There are also Supplementary, which include six tables and two figures. The Discussion is based on the data obtained. The list of references includes 89 sources. All of them are related to the presented work.

Notes.

The manuscript does not have a separate "Conclusion" section. I suggest that the authors consider creating a separate "Conclusion" section where the main findings could be summarized. This could be the last paragraph of the "Discussion" section, separated into the "Conclusion" section. - We elected not to use a Conclusion section because the Discussion is not unusually long or complex.

After minor editorial corrections and additions, the manuscript may be published in the Journal of Fungi, special issue “Diversity and Ecology of Fungi from Underexplored and Extreme Environments”. - Yes, this is where we intended to publish this paper.

Reviewer 3 Report

The manuscript “Xerophilic Aspergillaceae Dominate the Communities of Fungi Recovered from the Mound Nests of the Western Thatching Ant (Formica obscuripes)” is well written and organized, has a clear structure, and the results are well illustrated. In my humble opinion, the article is very interesting and important, both from a soil biodiversity and ecological point of view. Only a few minor points should be clarified.

Lines 86-87: “Non-mound soils (N) were collected 1 m SW of each mound to a depth of 3 cm”

Why was the soil sampled from a depth of 3 cm? Normally, when studying microbial communities in topsoil, the depth of the top layer 0-10/20 cm is considered. Please clarify.

https://doi.org/10.1016/j.soilbio.2013.02.006

https://doi.org/10.1016/j.geoderma.2019.114164

https://doi.org/10.1016/j.soilbio.2014.04.023

Line 116 “Inoculated plates were incubated at RT for 4-5 d after which colonies of filamentous “

It is better not to use RT, but to specify the incubation temperature for the plates.

Lines 346-348 “Mean growth rates of all taxa were higher on MYA containing an osmoticant, and most species grew more rapidly on MYA + 1 M sucrose (aw = 0.97) than on MYA + 1 M glycerol (aw = 0.97)”

It would be nice if the authors would at least discuss this aspect in a few sentences: why faster growth was observed in all taxa on a medium with sucrose.

The manuscript “Xerophilic Aspergillaceae Dominate the Communities of Fungi Recovered from the Mound Nests of the Western Thatching Ant (Formica obscuripes)” is well written and organized, has a clear structure, and the results are well illustrated. In my humble opinion, the article is very interesting and important, both from a soil biodiversity and ecological point of view. Only a few minor points should be clarified.

Lines 86-87: “Non-mound soils (N) were collected 1 m SW of each mound to a depth of 3 cm”

Why was the soil sampled from a depth of 3 cm? Normally, when studying microbial communities in topsoil, the depth of the top layer 0-10/20 cm is considered. Please clarify.

https://doi.org/10.1016/j.soilbio.2013.02.006

https://doi.org/10.1016/j.geoderma.2019.114164

https://doi.org/10.1016/j.soilbio.2014.04.023

Line 116 “Inoculated plates were incubated at RT for 4-5 d after which colonies of filamentous “

It is better not to use RT, but to specify the incubation temperature for the plates.

Lines 346-348 “Mean growth rates of all taxa were higher on MYA containing an osmoticant, and most species grew more rapidly on MYA + 1 M sucrose (aw = 0.97) than on MYA + 1 M glycerol (aw = 0.97)”

It would be nice if the authors would at least discuss this aspect in a few sentences: why faster growth was observed in all taxa on a medium with sucrose.

Author Response

Reviewer 3

 I would not like to sign my review report

 I would like to sign my review report

Does the title describe the article's topic with sufficient precision?

Yes

No

Does the introduction provide a comprehensive yet concise overview about the state of knowledge in the area of research?

Yes

No

Is the research design appropriate and are the methods adequately described?

Yes

No

Are the results presented clearly and in sufficient detail, are the conclusions supported by the results and are they put into context within the existing literature?

Yes

No

Are all of the cited references relevant to the research?

Yes

No

Does this article provide a relevant contribution to the scientific discussion of this topic?

Yes

No

English language and style

I am not qualified to assess the quality of English in this paper.

The English is very difficult to understand/incomprehensible.

Extensive editing of English language required.

Moderate editing of English language required.

Minor editing of English language required.

English language fine. No issues detected.

Comments for Authors

Advice for completing your review can be found at: https://www.mdpi.com/reviewers#Review_Report

Major comments

The manuscript “Xerophilic Aspergillaceae Dominate the Communities of Fungi Recovered from the Mound Nests of the Western Thatching Ant (Formica obscuripes)” is well written and organized, has a clear structure, and the results are well illustrated. In my humble opinion, the article is very interesting and important, both from a soil biodiversity and ecological point of view. Only a few minor points should be clarified.

Lines 86-87: “Non-mound soils (N) were collected 1 m SW of each mound to a depth of 3 cm”

Why was the soil sampled from a depth of 3 cm? Normally, when studying microbial communities in topsoil, the depth of the top layer 0-10/20 cm is considered. Please clarify. - We sampled non-mound soils to the same depth (3 cm) for comparative purposes (i.e., to be consistent with the depth to which we sampled soils from the tops of mounds). The depth to which soils from the tops of mounds were collected given as 5 cm in the submitted manuscript.  This error has been corrected.

https://doi.org/10.1016/j.soilbio.2013.02.006

https://doi.org/10.1016/j.geoderma.2019.114164

https://doi.org/10.1016/j.soilbio.2014.04.023

Line 116 “Inoculated plates were incubated at RT for 4-5 d after which colonies of filamentous “ It is better not to use RT, but to specify the incubation temperature for the plates. - We agree, but lacking sufficient incubator space, we elected to incubate all isolation plates and plates to which colonies were sub-cultured in one lab where temperatures varied only between 18-21 C.  This temperature range is described as room temperature (RT) in Section 2.1 of the Materials and Methods.

Lines 346-348 “Mean growth rates of all taxa were higher on MYA containing an osmoticant, and most species grew more rapidly on MYA + 1 M sucrose (aw = 0.97) than on MYA + 1 M glycerol (aw = 0.97)”

It would be nice if the authors would at least discuss this aspect in a few sentences: why faster growth was observed in all taxa on a medium with sucrose. - We speculate that the slightly faster growth rates of Fungi from ant nests on a medium containing 1 M sucrose reflects a preference for this osmoticant as a source of carbon and energy over glycerol.  However, we are reluctant to discuss this aspect of our results given that the growth rate study was designed only to assess the xerotolerances of the Fungi we isolated.

Reviewer 4 Report

The manuscript by Gross et al., investigated culturable fungal community in the mound nest soils of Formica obscuripes. The authors performed a lot of work on isolation and characterization of the fungal species from the soil samples, and the results indicated the impact of environmental factors (water activity and content) on the culturable fungal communities in the soils. I have several comments regarding the interpretation and discussion of the results as listed in the Detail comments.

Title. It needs to be specified that the “Communities of Fungi” is the culturable fungus in the mound nest soils, rather than the entire fungal community. As the vast majority fungal species in the soils are un-culturable, which is not investigated in this study.

Line 13. The “3594 isolates” needs to be more specific. i.e., these 3954 isolates belong to how many families/genus/species? Same for Table 4, 5. Such numbers are unclear as most of the isolates are actually the same/replicates.

The authors compared the culturable fungal community of the soils from the mound nests, within nests, and non-nest mounds sites (Fig 1). The figure shows the fungal communities in the soils of different sites (with different water conditions) are significant different. What’s the reason for this? This is the most important finding of this study and need to be more clearly and deeply discussed. 

Since the fungal species of this study were isolated using rich media, I was wondering the actual abundance of your isolates in the natural soils (not the relative abundance as compared to culturable community)? What’s the possible interaction between Formica obscuripes and your isolated fungal species? And what’s the possible ecological roles of such interaction? These also need to be further discussed in the Discussion section.

Author Response

Reviewer 4

 I would not like to sign my review report

 I would like to sign my review report

Does the title describe the article's topic with sufficient precision?

Yes

No

Title is unspecific that can be misleading - The title has been changed.

Does the introduction provide a comprehensive yet concise overview about the state of knowledge in the area of research?

Yes

No

Is the research design appropriate and are the methods adequately described?

Yes

No

Are the results presented clearly and in sufficient detail, are the conclusions supported by the results and are they put into context within the existing literature?

Yes

No

Are all of the cited references relevant to the research?

Yes

No

Does this article provide a relevant contribution to the scientific discussion of this topic?

Yes

No

English language and style

I am not qualified to assess the quality of English in this paper.

The English is very difficult to understand/incomprehensible.

Extensive editing of English language required.

Moderate editing of English language required.

Minor editing of English language required.

English language fine. No issues detected.

Comments for Authors

Advice for completing your review can be found at: https://www.mdpi.com/reviewers#Review_Report

Major comments

The manuscript by Gross et al., investigated culturable fungal community in the mound nest soils of Formica obscuripes. The authors performed a lot of work on isolation and characterization of the fungal species from the soil samples, and the results indicated the impact of environmental factors (water activity and content) on the culturable fungal communities in the soils. I have several comments regarding the interpretation and discussion of the results as listed in the Detail comments.

Detail comments

Title. It needs to be specified that the “Communities of Fungi” is the culturable fungus in the mound nest soils, rather than the entire fungal community. As the vast majority fungal species in the soils are un-culturable, which is not investigated in this study. - This is a good point. We have changed the title of the ms to “Xerophilic Aspergillaceae Dominate the Communities of Culturable Fungi from the Mound Nests of the Western Thatching Ant (Formica obscuripes)”.

Line 13. The “3594 isolates” needs to be more specific. i.e., these 3954 isolates belong to how many families/genus/species? Same for Table 4, 5. Such numbers are unclear as most of the isolates are actually the same/replicates. - We have now included the number of families of Ascomycota to which the 98 species are currently assigned.  The numbers of species, genera and families isolated on each of the three media (Table 4, 5) and are provided in Supplementary Tables 1-3.

The authors compared the culturable fungal community of the soils from the mound nests, within nests, and non-nest mounds sites (Fig 1). The figure shows the fungal communities in the soils of different sites (with different water conditions) are significant different. What’s the reason for this? This is the most important finding of this study and need to be more clearly and deeply discussed. - We do not know why communities of Fungi from mound tops, with mounds and from non-mound site are different.  The mounds nest of F. obscuripes, like those of many other species of ants, are unique, non-homogenous environments that support distinct communities of Fungi but we are unaware of any studies that demonstrate that these mycota are stable between seasons or from year to year.  We also have a limited understanding of how these mycota are influenced by the ants, other microbial and invertebrate co-inhabitants of mounds, and the communities of plants that surround each nest.  This is why ant nests and other animal-modified habitats must be studied more intensively using a variety of methods (cultural and molecular).

Since the fungal species of this study were isolated using rich media, I was wondering the actual abundance of your isolates in the natural soils (not the relative abundance as compared to culturable community)? What’s the possible interaction between Formica obscuripes and your isolated fungal species? And what’s the possible ecological roles of such interaction? These also need to be further discussed in the Discussion section. - We avoided discussing these (and other important questions) because they were not the focus of our study.